# Exploring the use of a digital therapeutic intervention to support the pediatric cardiac care journey: Qualitative study on clinician perspectives

**Sahr Wali**[1,2,3]*, **Alliya Remtulla Tharani**[4], **Diana Balmer-Minnes**[4], **Joseph A. Cafazzo**[1,2,5,6], **Jessica Laks**[7], **Aamir Jeewa**[4]

1 Institute of Health Policy, Management and Evaluation, University of Toronto, Toronto, Ontario, Canada,
2 Centre for Digital Therapeutics, University Health Network, Toronto, Ontario, Canada, 3 Ted Rogers Centre for Heart Research, University Health Network, Toronto, Ontario, Canada, 4 Labatt Family Heart Centre, Hospital for Sick Children, Toronto, Ontario, Canada, 5 Institute of Biomaterials and Biomedical Engineering, University of Toronto, Toronto, Ontario, Canada, 6 Department of Computer Science, University of Toronto, Toronto, Ontario, Canada, 7 Johns Hopkins All Children's Hospital, St Petersburg, Florida, United States of America

* sahr.wali@mail.utoronto.ca

**Data Availability Statement:** Data leading to this study findings is available without restriction within the manuscript and in our supplementary files.

## Abstract

Pediatric heart disease currently effects over one million infants, children, and adolescents in the United States alone. Unlike the adult population, pediatric patients face a more uncertain path with factors relating to their growth and maturation creating levels of complexity to their care management. With mobile phones increasingly being utilized amongst adolescents, digital therapeutics tools could provide a platform to help patients and families manage their condition. This study explored clinicians' views on the use of a digital therapeutic program to support pediatric heart disease management. Using the principles from user-centered design, semi-structured interviews were conducted with 4 cardiologists, 3 nurse practitioners and 1 cardiology fellow at the Hospital for Sick Children. All interview transcripts underwent inductive thematic analysis using Braun and Clarke's iterative six-phase approach. To further contextualize the analytic interpretation of the study findings, Eakin and Gladstone's value-adding approach was used. Five themes were identified: (i) multidisciplinary model of care; (ii) patient care needs change over time; (iii) treatment burden and difficulties in care management; (iv) transition to adulthood; and (v) filling care gaps with digital health. Clinicians valued the opportunity to monitor a patient's health status in real-time, as it allowed them to modify care regimens on a more preventive basis. However, with adolescent care often varying according to the patient's age and disease severity, a digital therapeutic program would only be valuable if it was customizable to the patients changing care journey. Digital therapeutic programs can ease the process of self-care for adolescents with heart disease throughout the growth and maturation of their care journey. However, to ensure the sustained use of a program, there is a need to work collaboratively with patients, caregivers, and clinicians to ensure their lived experiences guide the design and delivery of the overall program.

**Funding:** This work was supported by the Emmet Jeffrey Foster Endowment Fund (AJ). The funders had no role in study design, data collection and analysis, decision to publish, or preparation of the manuscript.

**Competing interests:** The authors have declared that no competing interests exist.

## Author summary

Digital therapeutics have become increasingly used as an evidence-based tool to support heart disease management, but limited evidence exists on its impact and value in the pediatric population. With factors related to growth and maturation creating levels of complexity to the pediatric care journey, digital therapeutics can offer a personalized avenue of support to meet the changing needs of patients and their families. In this study, we explored the clinician perspective on the current challenges associated with pediatric cardiac care, as well the opportunity for digital therapeutics to support patient management and clinical decision-making. We found that clinicians valued the availability of real-time data and remote support, but the digital therapeutic program would only be valuable if it was customizable to the patient's changes in lifestyle and disease severity. This work will help to identify the design requirements for co-design of a digital therapeutic program supporting pediatric heart disease management.

## Introduction

Pediatric heart disease currently represents a significant cause of morbidity and mortality in childhood, effecting over one million infants, children, and adolescents in the United States alone [1,2]. The complexities associated with pediatric heart disease has been connected to a diverse range of primary, secondary, and shared pathways of disease progression [3,4]. Among children with heart failure, disease processes such as primary cardiomyopathy, rhythm disorders, congenital heart disease (CHD), and acquired heart diseases (e.g., myocarditis, rhematic heart disease, Kawasaki disease) have been linked as the causes of the condition [5–7]. These sources for HF differ substantially from the adult population, leading children with HF-related hospitalizations to have over a 20-fold increased risk of death [3,7–11]. With the advancements in medical and surgical care, improvements in life expectancy have been reported amongst children with HF [1,5]. Traditionally, as part of the HF treatment regimen, children and their caregivers are expected to manage a series of self-care tasks, including medication, weight, blood pressure, diet and physical activity management [11,12]. Effective monitoring of their treatment regimen is pivotal to prevent the occurrence of disease progression or hospitalization and need to be tailored to the age of the child as presentations of HF varies considerably between infants and older children [1,11]. It is also important to recognize that the monitoring process can often be difficult to manage, especially for parents or other caregivers, who must oversee both the child's treatment regimen as well as other family responsibilities [1,12].

With the widespread penetration of mobile phones, there has been an increasing use of digital health tools to support patient care management [13,14]. Amongst the younger demographic, adolescents have already been reported to have one of the highest rates of mobile phone use, with over 60% having frequently downloaded mobile applications (apps) [13]. With this, several digital health interventions have been introduced in the clinical setting to provide patients with a source of education, remote monitoring, and disease management [13,14]. Notably, digital therapeutics (DTx) has rapidly emerged as an attractive care modality, as its use of evidence-based software has transformed the delivery of patient care [15–18]. The advantage of DTx over traditional approaches to care management, specifically among the younger, "tech-savvy" generation, involves its ability to provide an interactive and personalized platform to help patients and families monitor their care in the comfort of their own home [13,19,20]. Nevertheless, despite the benefits highlighted, the use of DTx amongst the pediatric

population remains to be in its infancy, as there is currently a limited number of studies investigating pediatric-focused DTx interventions [17,18]. Thus, with the difficulties associated with pediatric heart disease and the benefits of DTx, the objective of this study was to explore how adolescent heart disease care is currently being delivered and whether a DTx intervention can be utilized to better support patient, caregiver, and clinician needs.

## Methods

### Setting/intervention

The Peter Munk Cardiac Centre (PMCC) and the Centre for Digital Therapeutics at the University Health Network (UHN) in Toronto, Canada, have designed a mobile phone-based digital therapeutic, known as the *Medly* program (https://medly.ca), to provide remote clinical support for patients with HF [15,19]. *Medly* currently operates as part of the standard of care at the UHN Heart Function Clinic in Toronto, Canada [19]. With *Medly*, patients can use a mobile phone with the *Medly* app, weight scale, and blood pressure monitor to record daily physiologic readings and symptoms [19]. These readings run through a rules-based algorithm to automatically generate individualized self-care instructions to the patients, while simultaneously alerting clinicians when there are changes in the patient's health status beyond their set clinical parameters [19]. By empowering patients to take a more active role in their self-management, *Medly* has been shown to **reduce patient HF-related hospitalizations** by 50% and **all-cause hospitalizations by 24% [19].**

Currently, the *Medly* program solely serves an adult population, thus with the varying needs and care management plans amongst the pediatric population, the present design and delivery of *Medly* may not be appropriate or applicable for their use. To explore the potential value of integrating and/or adapt the *Medly* program for adolescent cardiac care, the PMCC partnered with the Labatt Family Heart Centre (LFHC) at the Hospital for Sick Children in Toronto, Canada.

### Study design

This qualitative study was designed to elicit insights from clinicians regarding the current care model for adolescents with heart disease. Guided by the principles of user-centered design (UCD), clinician feedback was used to help better understand the challenges associated with patient care, as well as how an existing DTx program could be used to help support symptom management and clinical decision-making [21,22].

### Recruitment of participants

Clinicians were purposefully identified and invited to participate in the study. Eligible participants consisted of clinicians (i.e., pediatric cardiologists, trainees, nurse practitioners, registered nurses) that were employed at the Hospital of Sick Children and were involved in the care of patients with i) HF due to primary cardiomyopathies or secondary to congenital heart disease, ii) congenital heart disease requiring frequent hospital visits (>3–4 visits/year), or iii) post-operative cardiac patients. All clinicians meeting the eligibility criteria were invited to participate in the study via an email invitation. Interviews were conducted until data saturation was reached, whereby no new perspectives or ideas were generated from the data [23]. This study was approved by the Hospital for Sick Kids Research Ethics Board (CTO: 1799) and the University Health Network Research Ethics Board (UHN: 19–6022).

### Data collection and analysis

Semi-structured interviews were conducted individually with each clinician, using an ethnographic research approach. Inteview guides were developed in collaboration with the PMCC and LFHC investigators to probe for a more detailed understanding of heart disease management. During each interview, clincians were asked to comment on their experiences managing patient care, as well as whether a program like *Medly* would be useful to help support clinical decision-making. Interviews were conducted between November 20th, 2019, to April 3rd, 2020, and lasted approximately 45–60 minutes. Written informed consent was obtained from all participants, whereby clinicians were aware that the interviews were audio recorded and subsequently transcribed verbatim for data analysis.

Using Braun and Clarke's iterative six-phase approach, interview transcripts underwent inductive thematic analysis [23]. Two authors were involved in the data analysis process (ART, SW). To improve the trustworthiness of the analysis, the authors engaged in both procedural and analytical memoing, where transcripts and analytic memos were entered into NVivo 12 (QSR International) [24]. The authors (ART, SW) also used Eakin and Gladstone's "value-adding" approach to qualitative analysis to enrich and contextualize their analytic interpretation of the study findings [25]. By using analytic devices such as reflexivity and generative coding to review the transcripts, the authors were able to deepen their interpretation of the data in more abstract terms. Initially, each author (ART, SW) independently coded the transcripts, then they collaboratively discussed the results and discrepancies in their findings. Following the initial data review, codes were grouped into categories to identity themes relating the research question. After a series of four analytic discussions, all authors from the research team met collectively to develop the study themes. In alignment with the principles of UCD, the final themes from this analysis aim to be used to help inform the design of the adapted pediatric-specific DTx system in a future study.

## Results

### Participant characteristics

A total of 8 participants were interviewed: 4 cardiologists, 3 nurse practitioners (NPs) and 1 cardiology fellow.

### Interview findings

During the analysis of the interview data, five themes were identified: (i) multidisciplinary model of care; (ii) patient care needs change over time; (iii) treatment burden and difficulties in care management; (iv) transition to adulthood; and (v) filling care gaps with digital health.

**(i) Multidiscplinary model of care.**   With the level of complexity associated with adolescent heart disease, a multidisciplinary approach to patient care was commonly utilized. Clinicians expressed that adolescent patients often faced a diverse range of health challenges that made the use of a more collaborative approach to care more desirable. The nature of care collaboration was also reported to go beyond just medication management, as components of nutrition and social support were incorporated to optimize patient well-being.

"*So, it's [patient care] a multidisciplinary approach. We have physicians, nurse practitioners, and then other staff that include a dietician, social worker, physical therapy, occupational therapy, among others. We will try to optimize their care from, not just from a medication standpoint, but from a nutrition standpoint, from a psychosocial aspect, and ensuring that they're getting good cardiac rehab, as much as possible*" (Clinician 4- Nurse Practitioner)

Although a multidisciplinary model of care was utilized, the patient's cardiologist remained the most responsible for their care. Various clinical tasks would be completed or facilitated by other members of the care team (i.e., pharmacist, family doctor, nurse), however, clinical decisions would primarily be managed by the patient's cardiologist.

*"As a physician I would say. I think we work closely with other physicians in our roles and we do make some adjustments but it's always kind of running it by the physicians as well [all members of the care team], making sure they agree or revising the plan if they see fit" (Clinician 5—Cardiologist)*

*"If we are making medication changes then they may need to have blood pressures, weights that type of stuff done as an outpatient and that might get done either in like a family doctor's office sometimes or like a pharmacy. But I would say the majority of the type of care and follow up and all of that stuff is probably managed like through the healthcare professionals [cardiologists]" (Clinician 1- Cardiologist)*

Sources of collaboration would often also be initiated based on the cardiologist's clinical decision or recommendation for a patient's treatment plan. For instance, in the case that clinicians felt that their patients required additional support for the various psychosocial factors impacting their well-being, a social worker would readily be referred to join their care team. In addition, as part of the multidisciplinary approach to care, when patients faced changes in their clinical status (i.e., severe HF), additional specialist support would be obtained by their clinicians.

*"I mean we [cardiologists] get social workers involved if we see there are any issues and need question about supports" [Clinician 1- Cardiologist]*

*"From a multidisciplinary approach, we do involve the pharmacist, the therapists, the dietician, social worker, on a much more frequent basis. Then, if there's any other kind of other organ involvement, whether it's hematological or neurological, or kidney related, we'll seek out those other support services as needed. We also involved cardiac surgery, in many of our patients, because some of these patients need either a surgical repair or a surgical optimization or, eventually, if they have end stage heart failure, may need surgery to help with the cardiac output." [Clinician 4- Nurse Practitioner]*

**(ii) Patient care needs change over time.**   Contrary to the adult population, clinicians reported that adolescent patients faced a significant variance in their needs and treatment plan based on their age, as well as their disease severity. Depending on these factors, the frequency and duration of patient follow-ups would differ, whereby, infants would generally require a more routine bi-weekly visit, whereas adolescents would be seen every three months to a year. However, it is important to note that if the patient's condition was more severe, regular weekly to monthly visits would be scheduled to monitor the patients' health status.

*"And it all depends; some patients will–and these are primarily the infants I would say. You know, we maybe see on a weekly basis or every two weeks routinely to make sure they're gaining weight and that they have good symptom management. For older kids it could be as often as once a month. I would say that's rare. But usually, it would be every three months and for the less acute more stable maybe once a year, once every six months" [Clinician 5- Cardiologist]*

"*If they're patients that are not in heart failure but have some sort of cardiomyopathy, if they have like a phenotype positive cardiomyopathy but are not in heart failure at the time, we would see them every year. If they're in heart failure and were actively making changes to medication or they're recently discharged from hospital, then we would see them more frequently* [Clinician 1- Cardiologist]

With the varying patient groups amongst the adolescent population, clinicians indicated that they most often assessed a patient's condition based on a series of metrics related to their vital signs, growth, nutrition and activity level. Factors relating to school performance and participation would also be considered in order to better assess whether the patient's condition was interfering with their daily life.

So, in terms or metrics vitals are very important, with that the biggest things would be heart rate, blood pressure, weight and weight changes. If they're seen more frequently, we may be looking at fluid balances as well [Clinician 1- Cardiologist]

"*I would say most commonly their activity level, the sports and things they're participating in, depending on how bad they are then their activities of daily living, number of medications and doses of medications that they're on, their symptoms and then their growth, weight gain. Sometimes in that age group you could maybe throw in school performance if there's a lot of school absentees and related to medical issues, those would probably be kind of the key things from clinic, from a clinic visit*" [Clinician 2- Cardiologist]

One clinician also highlighted that despite using the aforementioned clinical metrics, these core characteristics may be modified to better suit the condition and needs of each patient on an individual basis. Specifically, in the case of cardiomyopathy patients, reviewing individual heart rhythm was found to be more useful than blood pressure or heart rate.

"*Yeah, because if it's a cardiomyopathy patient, a hypertrophic cardiomyopathy patient, I don't care about their, maybe blood pressure, heart rate, but I may care more about their rhythm*" [Clinician 3- Cardiologist]

While considering the clinical metrics relevant to adolescent heart disease, active engagement both in terms of symptom management and daily living were frequently encouraged. Clinicians indicated a range of self-care related tasks that were pivotal to both the management of their condition, as well as their developmental progress. However, despite the importance of these tasks, the benefits of self-care were dependent on patient treatment compliance.

"*So, active engagement is what we always hope for. Whether it's, parent and patient together, or the patient themselves. We want them to take their take their medications as regularly as possible. We want them to try to adhere to recommendations with regards to physical activity and diet. We also want to know and listen to any kind of symptoms that are worsening from them and if they're able to complete all their activities of daily living, like going to school. Self-care, brushing teeth, showering, all those kinds of things. And that they're also making developmental progress as well too, that's age appropriate.*" [Clinician 7 –Nurse Practitioner]

"*We follow patients, and we give them instruction about what to watch for in terms of symptoms. If they're overweight or if they have issues with hypertension, we give them guidance of or some strategies around that. But then, it's mostly up to them unless they're the extreme.*

*And then when you follow them up you see how well they've done or if they were able to comply or not"* [Clinician 5- Cardiologist]

**(iii) Treatment burden and difficulties in care management.** With the number of tasks associated with patient care management, clinicians felt that patients were often overwhelmed by the treatment workload. Specifically, medication regimens that went beyond once a day were viewed as burdensome, since it interfered with other aspects of care management throughout the day (e.g., nutrition, activity). Other aspects of care that seemed simple in nature, such as travelling to the hospital for clinic visits, were very difficult for some patients and families depending on their geographic location and the frequency of their visits.

"*I think anytime you have medication regimens that start to go beyond once a day medicines, then I think that also becomes more difficult for families to manage. I think optimization of nutrition can also be quite stressful, especially if the child is failure to thrive or obese, either end of the spectrum, then I think managing that nutrition is also, can be, difficult for families"* [Clinician 4- Nurse Practitioner]

"*I think coming to the hospital it is tiring, and I think, so I think frequent visits to the hospital, for any reason, are always difficult and challenging. Especially if they live in further away places. Then, I think that becomes more burdensome to families* [Clinician 4- Nurse Practitioner]

For patients that were able to adequately manage their care regimen, it was found that some began to question the necessity of all the care tasks. Clinicians indicated that patients often confused their health with an indication that they no longer required treatment, when their well-being was rather a result of their treatment adherence.

"*If we have a well-managed heart failure patient on medication and everything, they often will start to feel–they feel good, they feel healthy. And so, then question may come up do I need these medications, or do I need to be getting blood work this often or do I need to be followed this often because I feel fine? When in reality they probably feel fine because they're taking all that stuff"* [Clinician 3- Cardiologist]

In reflection of the various patient age groups, one clinician also found that adolescents were less motivated to maintain their treatment regimen, leading to a higher rate of non-adherence. In some cases, adolescent patients were reported to avoid clinic appointments, as well as conceal the severity of their condition. This discrepancy in adherence was attributed due to a series of psychosocial factors (i.e., mood, relationships, social support), as patients were concerned on how their condition would impact their lifestyle outside of the home setting.

"*I think the non-adherence rate is higher in adolescents. I think there are some teenage groups that will try to avoid or hide coming to the doctor as much as possible, so they won't necessarily disclose all their symptoms very easily.–I think depending on the underlying cause of their heart failure, you know, failure to thrive is also a problem, as opposed to not just usually fluid overload that's often seen in the adult side of things. I think, with regards to adolescents, their psychosocial is affected quite a bit with regards to their mood and to how they're going to continue with their peers and school and their other relationships."* [Clinician 4- Nurse Practitioner]

**(iv) Transition to adulthood.** With patients continuing to grow throughout their care journey, clinicians reported that they would also be expected to take more control of their self-management tasks (e.g., managing symptoms, reporting changes in health to clinician). During the younger age range, patients were notably more reliant on their caregivers (i.e., parent, family) to handle both the clinical and administrative roles associated with their care management (e.g., administering medication, booking appointments). With this newfound independence, this dynamic would soon change, leading to either a greater sense of care ownership or non-compliance.

"*But over time, their model of care should change to be more self-management and more ownership of their medical condition especially in preparing for transition to the adult world*" [Clinician 1- Cardiologist]

"*I think they start off as younger age children and at their developmental level, they're more reliant on their parents. But as they get older and then their needs may be less dependent on their parents for administration of medications or for managing their activities of daily living. So, it becomes dealing with their burgeoning independence more*" [Clinician 4- Nurse Practitioner]

"As they start to get older and more independent, then there's the concept of them taking on more ownership for their care and being a little more self-reliant. But it also, then, has a flip side of them potentially being non-adherent and having poor compliance with regards to their healthcare." [Clinician 2- Cardiologist]

Unique to the Hospital for Sick Children, clinicians reported that an adolescent medicine team was available to help support patients during the transition to adulthood. This team of clinicians were noted to help with a range of both clinical and personal issues patients were facing (e.g., sex, drugs, complex care). With these resources in place, patients were able to obtain a better sense of guidance for health issues unique to their age range.

"*So adolescent medicine is a physician team, physician group, and nurse practitioners and all of that that their special–that they specialize in exactly that, adolescent health. So, they deal a lot with like anything that the adolescents are going through like sex, drugs, coping mechanisms and especially they have expertise in the area of dealing with adolescent who are also going through like complex medical conditions and complex care and dealing with that type of stuff.*" [Clinician 1- Cardiologist]

While recognizing the importance and difficulties associated with the growth to adulthood, during the actual transition process to adult services, patients were often given a referral and expected to restart their care journey. Specifically, clinicians reported that in most cases, the transition process simply involved summarizing the patient's information and forwarding it to a clinic closest to their home. With this, despite having several care resources throughout the adolescent period, there was a lack of clarity on what the transition to adult services would look like, and concern on whether the care provided at adult institutions would be at the same caliber as the care in pediatrics.

"*So, it's nothing formalized in the sense that it's–they come for a last visit, their data gets summarized, they sign consent and then we send the referrals to the place that's closest to their home* [Clinician 2- Cardiologist]

"If they are true heart function patients, they would get transitioned over to Toronto General, there would be a release of information that we would fill out and we would forward, send that information to the heart function team over there. In terms of having like a more thorough or more like sort of program in terms of transition I don't believe we have like an actual transition program" [Clinician 1- Cardiologist]

**(v) Filling care gaps with digital health.** With the challenges associated with care management, the use of a digital tool was viewed as a valuable opportunity to help support patients with their care routine, and clinicians with their clinical decision-making. Specifically, by allowing for remote symptom monitoring, some clinicians indicated that the use of real-time data would allow them to modify patient care regimens on a more preventive basis (e.g., medication doses, clinic visits needed). Patients living in more remote regions were also noted to significantly benefit from the digital tool, as it would reduce the need for travel, while providing updates on the patient's condition. Nevertheless, other clinicians felt wary about the benefits of a digital tool, as the clinical capabilities of the technology were uncertain (i.e., solely detect vital signs).

"–The value is going to be for patients who have symptoms that can be monitored, are on medications that could be adjusted or titrated and maybe–and there'd be an added benefit for people who are further away, right? So, you could modify a trip to the hospital or make better decisions about when they should come to the hospital to be assessed versus continuing interventions and observations in the home with reporting through the app" [Clinician 2]

"Oh yeah, absolutely. Especially we have patients that live in Sudbury and Thunder Bay, so that would certainly be very helpful for those patients. Oh, and also different provinces. We have patients in Nova Scotia and New Brunswick, so–Manitoba. So that's going to be very helpful" [Clinician 4]

"I'm curious whether this would help in avoiding some ER visits or not... If you knew how a patient was doing, based on their vitals, vitals are only good for certain [conditions]–and sometimes you do have to see a patient to know how well they're doing. The app is not going to measure your liver size or your JVP. So, it might give you a heart rate/blood pressure, but if it turns out that it can, if it does help us keep patients out of hospital, then that's a–even out of ER's, that would be of benefit" [Clinician 3]

The process of virtual decision-making was also highlighted as an area of concern, as clinicians often made key patient assessments during in-person clinic visits. To better support the effectiveness of a digital tool, clinicians expressed that data management and analytic processes, respective to each patient age range, should be incorporated within the technology. Specifically, the use of patient trends within the system was strongly desired by clinicians to help visualize clinical data (e.g., vital signs, weight). Ultimately, through this feature clinicians would gain a better sense of patient well-being without the need for a thorough clinical evaluation.

"Not having the patient in front of you to make some decisions is some of the challenges. It's always easier to bring a patient to clinic and see them, and lay hands on them. So, I think that's one of the challenges, is trying to come under grasp of doing things from, without having the patient right in front of you. I think some of it may be just navigating the technology itself and then coming up with any kind of pitfalls with regards to any gaps in the data or not

*getting perfect data, and then recognizing that data can be different for pediatric patients then for adult patients"* [Clinician 4]

*"Something that's kind of easy to read and understand. I'd like to see trend data with regards to some of the vital sign parameters. Like heart rate, respiratory rate, if possible, or blood pressure. I'd like to know their weights, also on trend basis, and then I think from a symptom's standpoint, I think some way to kind of get an easy-to-understand metric of, or score of what a patient feels like. Whether they're on the better end or the worse end of a lot of their symptoms"* [Clinician 4]

With the varying conditions associated with adolescents with heart disease, the metrics directing the functionality of the digital tool would need to be individualized according to both disease severity and patient age range. For instance, clinicians indicated that a guideline or standard set of core features could be used for all patients, but the program components would need to be further tailored for patients with specific cardiac conditions (e.g., monitor heart rate for cardiomyopathy patients). The frequency of tool interaction would also need to be modified in accordance with patient's treatment plan (i.e., daily or weekly symptom reporting), as many patients are not required to complete self-management tasks on a daily basis.

*"I think there'll be some that will be fairly standard across the board. But I think it would be nice to have a platform that is a little bit nimble as well too. So that maybe there's some tailoring, and maybe more for underlying heart failure disease spectrum, so kids who have certain types of cardiomyopathies, we can be a bit more tailored or precise for their problems, versus others"* [Clinician 4]

*"So maybe you're going to tell them every Monday or every Saturday check your weight and put in this and that, you know, heart rate, blood pressure, whatever we decide the demographics are going to be. We have very few patients that have any utility at all, a daily weight or a daily vital check. . . So, I would find it very hard to think of a significant portion of patients that need it done every day"* [Clinician 2]

Under the reality of adolescent care behaviours, clinicians expressed that to promote digital tool adoption, building incentive for patient usage would be crucial. Regardless of the type of features (i.e., gamification or rewards) included within the program, patients would need to find meaningful utility in the system to encourage them in their care management. With this, clinicians suggested that by providing a source of direct feedback from the clinician to the patient, this would motivate patients to better manage their care. Aside from the clinical feedback, to further incentivize program use, the design of the system would also need to be tailored to reflect the patient's unique preferences and capabilities (e.g., use of simple language, multiple language options, modifiable icon size, colors).

*"= There's no way a teenager's going to get up and consistently for the rest of the five years that they're at Sick Kids put in something every morning. It's just not going to happen. Even if you make it a game it doesn't matter how fun you make it, they're not going to do it. So, there has to be come utility or a trigger for it. . . I think if, form a tech standpoint, if it's cumbersome, if it's too slow, if it, you know, unstable, either that it crashes or things like that, I think that will make it less palatable to keep using it. I think if there isn't any kind of meaningful feedback from the clinician side of things, then they may start to lose some initiative [and] adherence to it as well"* [Clinician 2]

*"And then of course, there can be language barriers as well. So certainly, having the ability to use symbols or icons, that might actually be helpful for some of our families to help minimise that as a barrier"* [Clinician 8]

To sustain patient usage, clinician's felt that the incorporation of a reminder system would be fundamental to support care management regimens. However, in alignment with the need for customized program requirements (i.e., frequency of interactions), the level of patient reminders would also need to follow the same clinically guided principles. Some clinicians felt that to support patient usage, the digital tool could be linked to the caregiver's phone. This linkage would allow the caregiver to monitor the patients' health status and intervene with care reminders, as needed.

*"Ideally the system should recognize and send them a reminder, that's the ideal. And then maybe after two or three–you'd have to decide if it's weekly then you're going to have to know if half a week or if it's–right, that they'd missed it and they'd get a couple of reminders and then a phone call"* [Clinician 2]

*"The adolescent could have it on their phone as well, but it's linked to the app with the parents so that if the adolescent isn't entering the information that day the parent can see and they can either remind them or enter it for them, some sort of linking."* [Clinician 1]

Moreover, with patients gaining a greater level of care independence with age, caregiver access to the digital tool would need to be conditional based on patient preferences. Clinicians expressed that often younger patients were more reliant on caregiver support; thus, they were more likely to prefer the option for a shared platform to assist with their treatment regimen. Conversely, patients closer to adulthood were more responsible in their care management, leaving less incentive for further caregiver involvement. To ensure the privacy preferences of the patient are taken into consideration at all ages, the digital tool would need to provide modifiable access for the caregivers as they grow throughout their care journey

*"I can also see instances where an adolescent doesn't want their parent to know, which I think then we can be respectful for as well. I think that will just depend on the maturity level of the adolescents though. One [patient] that' s closer to 18, may want to be more responsible for their care. So, then they don't necessarily want to share those things with a caregiver, versus one who maybe at a younger age that does want more involvement and was willing to share all those things"* [Clinician 4]

*"Yeah, and for those that the teenagers who, you know, anyone 14 and up can ask us questions without their parents being present, right? Sometimes they need that privacy with us and so if that's even a possibility. Because I'm presuming the difference with adults and paediatrics Medly would be the parents being able to have access as well and seeing this, right? So, I don't know if it's already been considered, but something as the child gets older the teens, maybe like a private section with the teenager healthcare provider?"* [Clinician 8]

## Discussion

### Principal findings

With the complexities of pediatric heart disease continuing to grow as children move from infancy to adolescence, integrated solutions are needed to reduce the burden of care on

patients, families, and clinicians. In the adult population, digital health interventions have been widely introduced to empower remote heart disease management, however, there is minimal evidence regarding its use in the changing context of pediatric care [26,27]. With this, our study aimed to explore the clinician perspective on the applicability of DTx for pediatric heart disease. We found that despite patients facing varying needs and challenges according to their age and disease severity, the use of a DTx intervention was viewed as a valuable avenue to help support clinical decision-making and remote symptom monitoring. However, to support the patient care journey and sustain the value of the DTx program, various specifications would need to be integrated within the current multidisciplinary model of care.

Specifically, while the inclusion of a collaborative care approach was reported amongst clinicians, the facilitation and monitoring of these collaborative efforts would often lay at the direction of the cardiologist. This hierarchy of responsibility was found to add to the cardiologists already heavy workload, as there was an inherent need to oversee any updates to the patient's condition or care coordination. With this, clinicians indicated that the use of a DTx intervention could help to reduce the challenges associated with managing patient progress and clinical workflows, however, the process of virtual decision-making would need to be streamlined. In previous studies, digital tool-related workload burden often occurred due to staffing constraints and information overload caused by poor intervention implementation obstructing standard workflows [26,28]. Our findings suggest by integrating intervention-tailored workflows and trend-focused analytics (i.e., clinician dashboard), this will help provide a real-time assessment of a patient's well-being without burdening the cardiologists' workload [19,29,30].

Beyond clinical workflows, it is important to recognize that as patients move through the stages of youth to adolescence, the design of a DTx intervention would need to accommodate for both the patients' age- and disease-specific needs [1,3]. In many cases, interventions have been developed without considering the changing circumstances of a patients care journey, leaving systems to no longer be beneficial for their use [27]. By integrating individualized care plans within a digital health program, clinicians highlighted that this would provide patients with a more personal support system for their care. Recent studies have reported that the burden of care management often escalates for patients when there are changes to their care routine (i.e., new medication, symptom monitoring) [27,31,32]. With these challenges, to better support patients in managing their condition, intervention features would need to be modifiable to adjust for changes in the patient's clinical metrics (i.e., weight and blood pressure for HF, heart rhythm for cardiomyopathies), medications (i.e., type and dose), and frequency of interactions (i.e., daily, bi-weekly, or monthly symptom reporting/system reminders) required. Other metrics related to nutrition, activity level and cognitive performance should also be monitored to provide a more holistic assessment of the patient's condition and growth. Specifically, given that children with CHD are at an increased risk of developmental and cognitive abnormalities, factors contributing to neurocognitive impairment should be investigated, as it directly relates to a child's school performance and health-related quality of life [33]. In alignment with the pediatric care journey, it is important to recognize that despite patients gaining an increase in self-care independence with age, this also translated into a higher reporting of non-adherence. To better support active care management amongst the older pediatric demographic, clinicians indicated that interventions would need to provide a level of meaningful utility with its use. Our findings suggest by integrating direct sources of clinical guidance (i.e., self-care instructions, simplified education) within an intervention's functionality, this would motivate patients in their care management, as it is often difficult to obtain validated information on their specific condition [26]. Within the adolescent population, non-adherence has also been connected to a series of patient and family concerns regarding the transition from

child to adult services [27,34]. Despite the desire for care continuity, adolescents have been found to often be lost to follow-up after leaving pediatric care, as there was a level of uncertainty regarding the transfer process and whether adult services would provide the same quality of care [34]. Based on the recommendations from previous research, we argue by integrating transition plans within the intervention or program design, this would help incentivize self-care behaviours and overall intervention use [26,34]. Specifically, to build meaningful intervention utility, transition plans should aim to gradually introduce the expectations of adult services at an earlier stage and continue to work with patients to identify their preferences regarding caregiver involvement (i.e., supportive roles, privacy/access to patient data) throughout their care journey.

Despite the benefits of DTx for pediatric care, an unintended consequence of implementing digital health interventions involves its implications widening health disparities related to the digital divide [26,35,36]. By reporting symptoms in the home setting, DTx would provide an avenue for patients residing in remote communities to obtain a level of clinical guidance that was previously unavailable. However, with various geographic and socioeconomic barriers effecting patient families, the benefits of DTx solely apply to families able to obtain and afford the technical resources (i.e., Internet connectivity, mobile phone access, electricity) to support intervention use [26,35]. To help mitigate the barriers contributing to inequitable access to care, growing evidence indicates that interventions need to be co-designed directly with population groups to ensure different lenses of patient context are incorporated (i.e., low-income, ethnic minority) [35–37]. Specifically, by understanding the strength-based significance of various patient backgrounds (i.e., culture, history, language) and the limitations posed by the conditions of their social determinants of health (i.e., housing, income, race), we argue this would help to develop more patient-informed features within an intervention's design (i.e., offline features, language customizations).

## Limitations

This study was subject to several limitations. First, despite our aim to interview clinicians from a range of pediatric heart disease specialties, we were unable to recruit clinicians with specific expertise in all types of pediatric heart disease (such as arrhythmias or inflammatory conditions). Second, we did not interview registered nurses, which may have limited the level of operational insights on patient care management or clinical workflows. Despite our intent to obtain feedback on the use of DTx without implicit bias, one clinician was previously aware of the *Medly* Program operating at the Toronto General hospital. We recognize that these implications may have resulted in influencing participant feedback, leading our findings to be limited in its representation of the broader pediatric heart disease care experiences.

## Conclusion

As the number of children facing the burden of pediatric heart disease continues to grow, the level of innovative opportunities to improve patient quality of life remains promising. Unlike the adult population, pediatric patients face a more uncertain path with factors relating to their growth and maturation creating levels of complexity to their care management. Despite the limited evidence on DTx in the pediatric setting, its ability to support remote, collaborative sources of care continuity led clinicians of varying cardiac backgrounds to support its use. However, given the uniqueness of each patient's care journey, interventions should be co-designed with families to ensure both clinical and cultural factors are incorporated within its design.

## Supporting information

**S1 Text. Interview Guide: Clinician discussion guide on heart failure management and use of technology for care.**
(DOCX)

## Author Contributions

**Conceptualization:** Sahr Wali, Alliya Remtulla Tharani, Diana Balmer-Minnes, Joseph A. Cafazzo, Aamir Jeewa.

**Data curation:** Sahr Wali, Alliya Remtulla Tharani, Aamir Jeewa.

**Formal analysis:** Sahr Wali, Alliya Remtulla Tharani, Diana Balmer-Minnes.

**Funding acquisition:** Aamir Jeewa.

**Investigation:** Sahr Wali, Jessica Laks, Aamir Jeewa.

**Methodology:** Sahr Wali, Alliya Remtulla Tharani, Diana Balmer-Minnes, Jessica Laks, Aamir Jeewa.

**Project administration:** Sahr Wali, Diana Balmer-Minnes, Joseph A. Cafazzo, Jessica Laks, Aamir Jeewa.

**Resources:** Joseph A. Cafazzo.

**Software:** Joseph A. Cafazzo.

**Supervision:** Sahr Wali, Joseph A. Cafazzo, Aamir Jeewa.

**Validation:** Joseph A. Cafazzo, Aamir Jeewa.

**Visualization:** Sahr Wali.

**Writing – original draft:** Sahr Wali.

**Writing – review & editing:** Sahr Wali, Joseph A. Cafazzo, Jessica Laks, Aamir Jeewa.

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
