## [Decision Letter · Decision Letter 0]

25 Apr 2023

PDIG-D-23-00021

Exploring the Use of a Digital Therapeutic Intervention to Support the Pediatric Cardiac Care Journey: Qualitative Study on Clinician Perspectives

PLOS Digital Health

Dear Dr. Wali,

Thank you for submitting your manuscript to PLOS Digital Health. After careful consideration, we feel that it has merit but does not fully meet PLOS Digital Health's publication criteria as it currently stands. Therefore, we invite you to submit a revised version of the manuscript that addresses the points raised during the review process.

Please submit your revised manuscript within 60 days Jun 24 2023 11:59PM. If you will need more time than this to complete your revisions, please reply to this message or contact the journal office at digitalhealth@plos.org. Please include the following items when submitting your revised manuscript:

We look forward to receiving your revised manuscript.

Kind regards,

Jessica Keim-Malpass

Academic Editor

PLOS Digital Health

Journal Requirements:

Additional Editor Comments (if provided):

Reviewers' comments:

Reviewer's Responses to Questions

**Comments to the Author**

1. Does this manuscript meet PLOS Digital Health’s publication criteria? Is the manuscript technically sound, and do the data support the conclusions? The manuscript must describe methodologically and ethically rigorous research with conclusions that are appropriately drawn based on the data presented.

Reviewer #1: Yes

Reviewer #2: Yes

2. Has the statistical analysis been performed appropriately and rigorously?

Reviewer #1: N/A

Reviewer #2: N/A

3. Have the authors made all data underlying the findings in their manuscript fully available (please refer to the Data Availability Statement at the start of the manuscript PDF file)?

Reviewer #1: No

Reviewer #2: Yes

4. Is the manuscript presented in an intelligible fashion and written in standard English?

Reviewer #1: Yes

Reviewer #2: Yes

5. Review Comments to the Author

Reviewer #1: This is a formal review of manuscript number PDIG-D-23-00021 entitled "Exploring the Use of a Digital Therapeutic Intervention to Support the Pediatric Cardiac Care Journey: Qualitative Study on Clinician Perspectives." In this manuscript, the group highlights the expanding digital therapeutics in medicine while also highlighting the limited number of studies studying pediatric focused interventions. The aim of the study was to obtain baseline data regarding how digital therapeutics were currently being used as a way to understand how to better use these resources in the future.

Specific Comments:

1. Throughout the paper, the authors refer to their institutionalized program known as Medley. Sometimes when they refer to it, they italicize it, and other times they do not. The authors should be consistent throughout the paper.

2. The first section in the method section discusses the setting/intervention. At the end of this section, the authors discuss the study aim specifically related to how they will assess their Medley program. These paragraphs would be better suited in the introduction section. However, the introduction section was already at an appropriate length, so I encourage the authors to streamline it.

3. In the recruitment of participant section, line 142, the authors report congenital heart disease requiring frequent hospital visits. It would be helpful to know what the authors defined as frequent.

4. The authors provided a very detailed data collection and analysis which I found extremely interesting and could be expanded. Would it be possible for the authors to include the interview questions? It would also be interesting to include the response scoring and show how many times the five themes that were reported were identified, etc.

5. Table 1 does not add much. If the authors could expand more on how many years of practice each clinician had or some other demographic details, it may make the table more interesting.

6. It would also be helpful to know why the authors were only able to get eight people to consent for the study.

7. In the results section, I fear that the direct quotes, change the tone of the paper from more academic to read more like a news article or journalism article as opposed to an academic scientific paper. I am not suggesting removing them completely, but perhaps paraphrasing them to support what the authors' point is. This also made the results section extremely lengthy, and I think many readers would skip down to the discussion section.

Reviewer #2: PDIG-D-23-00021 Review

Major Comments

1. The Introduction focuses largely on congenital heart disease even though more than half of heart failure in adolescents is secondary to acquired heart disease. The third sentence of the first paragraph of the Introduction is very confusing as it suggests that patients with CHD are at risk for heart failure due to cardiomyopathy, myocarditis and other inflammatory or infectious etiologies as these are typically causes of acquired heart disease leading to heart failure. The Authors need to make a very clear distinction in the Introduction between heart failure due to acquired heart disease and heart failure due to congenital heart disease. Additionally, the Authors should confirm that the statistics noted are representative of pediatric heart failure in general and not just secondary to CHD.

2. One other area for exploration in the consideration of adoption of digital health technology in adolescents with heart failure secondary to congenital heart disease relates to the potential for neurocognitive deficits. These neurocognitive deficits may be associated with a history of palliated cyanotic heart disease (e.g., HLHS) or in patients with trisomy 21, which studies have demonstrated a high rate of heart failure in adult survivors with CHD. This is hinted at in the discussion related to school performance (ii. Patient care needs change over time) but a more nuanced discussion may be helpful.

Minor Comments

1. Abstract, Results – It is stated that “four themes were identified” though five themes are listed. Within the body of the manuscript, it is correctly stated to be “five themes”.

2. Results, Multidisciplinary model of care section, paragraph 1, sentence 2 – “patient” should be “patients”.

6. PLOS authors have the option to publish the peer review history of their article (what does this mean?). If published, this will include your full peer review and any attached files.

**Do you want your identity to be public for this peer review?** For information about this choice, including consent withdrawal, please see our Privacy Policy.

Reviewer #1: No

Reviewer #2: No

---

## [Decision Letter · Decision Letter 1]

19 Sep 2023

Exploring the Use of a Digital Therapeutic Intervention to Support the Pediatric Cardiac Care Journey: Qualitative Study on Clinician Perspectives

PDIG-D-23-00021R1

Dear Scientific Associate Wali,

We are pleased to inform you that your manuscript 'Exploring the Use of a Digital Therapeutic Intervention to Support the Pediatric Cardiac Care Journey: Qualitative Study on Clinician Perspectives' has been provisionally accepted for publication in PLOS Digital Health.

Best regards,

Padmanesan Narasimhan, MBBS MPH PhD

Section Editor

PLOS Digital Health

Reviewer Comments (if any, and for reference):

Reviewer's Responses to Questions

**Comments to the Author**

1. If the authors have adequately addressed your comments raised in a previous round of review and you feel that this manuscript is now acceptable for publication, you may indicate that here to bypass the “Comments to the Author” section, enter your conflict of interest statement in the “Confidential to Editor” section, and submit your "Accept" recommendation.

Reviewer #1: All comments have been addressed

Reviewer #2: All comments have been addressed

2. Does this manuscript meet PLOS Digital Health’s publication criteria? Is the manuscript technically sound, and do the data support the conclusions? The manuscript must describe methodologically and ethically rigorous research with conclusions that are appropriately drawn based on the data presented.

Reviewer #1: Yes

Reviewer #2: Yes

3. Has the statistical analysis been performed appropriately and rigorously?

Reviewer #1: N/A

Reviewer #2: Yes

4. Have the authors made all data underlying the findings in their manuscript fully available (please refer to the Data Availability Statement at the start of the manuscript PDF file)?

Reviewer #1: Yes

Reviewer #2: Yes

5. Is the manuscript presented in an intelligible fashion and written in standard English?

Reviewer #1: Yes

Reviewer #2: Yes

6. Review Comments to the Author

Reviewer #1: The authors have adequately addressed this reviewers comments and concerns.

Reviewer #2: The Authors have addressed all of my comments to my satisfaction and have revised the manuscript as appropriate.

7. PLOS authors have the option to publish the peer review history of their article (what does this mean?). If published, this will include your full peer review and any attached files.

**Do you want your identity to be public for this peer review?** For information about this choice, including consent withdrawal, please see our Privacy Policy.

Reviewer #1: No

Reviewer #2: **Yes: **Michael C. Spaeder
